# A Framework for Double-Blind Federated Adaptation of Foundation Models

**Nurbek Tastan[1], Karthik Nandakumar[1,2]**
[1]Mohamed bin Zayed University of Artificial Intelligence (MBZUAI)
[2]Michigan State University (MSU)
`nurbek.tastan@mbzuai.ac.ae, nandakum@msu.edu`

## Abstract

The availability of foundational models (FMs) pre-trained on large-scale data has advanced the state-of-the-art in many computer vision tasks. While FMs have demonstrated good zero-shot performance on many image classification tasks, there is often scope for performance improvement by adapting the FM to the downstream task. However, the data that is required for this adaptation typically exists in silos across multiple entities (data owners) and cannot be collated at a central location due to regulations and privacy concerns. At the same time, a learning service provider (LSP) who owns the FM cannot share the model with the data owners due to proprietary reasons. In some cases, the data owners may not have the resources to even store such large FMs. Hence, there is a need for algorithms to **adapt the FM in a double-blind federated manner**, i.e., the data owners do not know the FM or each other's data and the LSP does not see the data for the downstream tasks. In this work, we propose a framework for double-blind federated adaptation of FMs using fully homomorphic encryption (FHE). The proposed framework first decomposes the FM into a sequence of FHE-friendly blocks through knowledge distillation. The resulting FHE-friendly model is adapted for the downstream task via low-rank parallel adapters that can be learned without backpropagation through the FM. Since the proposed framework requires the LSP to share intermediate representations with the data owners, we design a privacy-preserving permutation scheme to prevent the data owners from learning the FM through model extraction attacks. Finally, a secure aggregation protocol is employed for federated learning of the low-rank parallel adapters. Empirical results on four datasets demonstrate the practical feasibility of the proposed framework.

## 1 Introduction

In the dynamic world of artificial intelligence (AI), the advent of foundational models (FMs) has marked a transformative era. Characterized by their vast scale, extensive pre-training, and versatility, FMs such as GPT (Radford et al., 2019; Brown et al., 2020), CLIP (Radford et al., 2021), BERT (Devlin et al., 2018; He et al., 2020; Liu et al., 2019), Stable Diffusion (Rombach et al., 2022), Segment Anything (Kirillov et al., 2023), and Vision Transformers (Dosovitskiy et al., 2020; Liu et al., 2021) have advanced the state-of-the-art (SOTA) in many machine learning, computer vision, and language processing tasks. Though vision and multimodal FMs have demonstrated good zero-shot performance on many image classification tasks, there is often scope for performance improvement when faced with challenging out-of-domain tasks (e.g., medical images or satellite imagery). Hence, it becomes essential to adapt the FM for the downstream task.

Adaptation of FMs for downstream tasks involves two main challenges - *computational complexity* and *data availability*. The simplest way to adapt an FM for a downstream classification task is via *transfer learning* - treat the FM as a feature extractor, append a classification head, and learn only the parameters of this classifier using the available training data (while the FM parameters remain frozen). If the classification head involves a single linear layer, it is referred to as *linear probing*. It is also possible to perform *partial* (only a selected subset of parameters are adapted) or *full finetuning* of the parameters of the FM based on the downstream data. Recently, two other broad categories of parameter-efficient fine-tuning (PEFT) approaches have been proposed for FM adaptation. The

first approach is called *prompt learning* (Jia et al., 2022), where input (or intermediate) prompts are learned instead of modifying the FM parameters. The second approach is referred to as *adapters* (Hu et al., 2021; Dettmers et al., 2023; Mercea et al., 2024), where additional components are added to the FM model architecture, and only these new parameters are learned. Adapters can be further categorized into sequential (e.g., low-rank adaptation a.k.a. LoRA (Hu et al., 2021)) and parallel (e.g., low-rank side adapter a.k.a. LoSA (Mercea et al., 2024)) adapters. Except for transfer learning and parallel adapters, all the other adaptation techniques require partial or complete backpropagation of the gradients through the FM, which is computationally expensive. Finally, when only query access to a black-box FM is allowed, gradients can be estimated through *zeroth-order optimization* (ZOO). While ZOO avoids backpropagation, it is also computationally expensive because it requires numerous forward passes.

The other major challenge in FM adaptation is that downstream training data required for adaptation may not be available to the learning service provider (LSP) who owns the FM. Moreover, this data may be distributed across multiple data owners (e.g., multiple hospitals or banks) and cannot be collated due to privacy concerns and regulations. This necessitates collaboration between the LSP and data owners to enable FM adaptation. Federated Learning (FL) (McMahan et al., 2017) is a paradigm that promises to alleviate this data availability challenge by facilitating collaborative model training across distributed data repositories, while maintaining the privacy of the underlying data. FL encompasses a spectrum of applications, from healthcare (Antunes et al., 2022; Xu et al., 2021) and finance (Long et al., 2020; Liu et al., 2023) to video surveillance (Sugianto et al., 2024) and beyond (Ghimire & Rawat, 2022). It has two primary scenarios: cross-silo (few data owners) and cross-device FL (large number of data owners) (Kairouz et al., 2021), each catering to different scales of collaboration, thereby enabling both large and small-scale entities to benefit.

In this work, we focus on the problem of federated adaptation of a foundation model (for an out-of-domain downstream image classification task) by an LSP (server) through collaboration with multiple data owners (clients) under two core constraints: (i) *Model privacy* - the LSP wants to retain full ownership of the FM and does not want to share the FM with the data owners; and (ii) *Data privacy* - the data owners do not want to leak their data to the LSP or other data owners. We jointly refer to these constraints as *double-blind privacy*. We make the following contributions:

- We propose a framework for double-blind federated adaptation of FMs based on well-known cryptographic constructs such as fully homomorphic encryption (FHE) and secure multi-party computation (MPC).
- The proposed framework first distills the knowledge from the FM into a sequence of FHE-friendly transformer attention blocks. It then employs an interactive protocol based on a privacy-preserving permutation scheme to perform encrypted inference without leaking the model and the data. It also leverages a low-rank parallel adapter for local learning and an MPC protocol for FL aggregation.
- We demonstrate the practical feasibility of the proposed framework through experiments on four datasets.

## 2 PROBLEM FORMULATION

Suppose that a foundation model (FM) $\mathcal{M}_\psi$ that is already pre-trained on a large-scale dataset is available with the learning service provider (LSP). The LSP aims to collaborate with the $K$ data owners to adapt the FM for a downstream image classification task. Each data owner $\mathcal{P}_k$ has access to a local training dataset $\mathcal{D}_k = \{\mathbf{x}_i^k, y_i^k\}_{i=1}^{N_k}$ corresponding to the downstream task. Here, $\mathbf{x}_i^k$ denotes the $i^{\text{th}}$ input image of $\mathcal{P}_k$, $y_i^k$ is the corresponding class label, $N_k$ is the number of training samples with $\mathcal{P}_k$, and $k \in [1, K]$.

**Problem Statement**: A task-specific classification head $\mathcal{H}_\eta$ and a side adapter $\mathcal{A}_\theta$ are appended to the FM to enable the adaptation. Thus, the adapted model $\widetilde{\mathcal{M}}_{\widetilde{\psi}}$ can be considered a combination of the original FM, the side adapter, and the classifier, i.e., $\widetilde{\mathcal{M}}_{\widetilde{\psi}} = (\mathcal{M}_\psi || \mathcal{A}_\theta) \circ \mathcal{H}_\eta$, where $||$ indicates that these functions operate in parallel and $\circ$ is the composition operator. The goal of the double-blind federated adaptation framework is to collaboratively learn the parameters $\eta$ and $\theta$ under the following constraints: (i) *Data Privacy*: the LSP does not learn anything about the local datasets

$\{\mathcal{D}_k\}_{k=1}^K$ and data owner $\mathcal{P}_k$ does not learn anything about other local datasets $\mathcal{D}_j$, where $j \neq k$; (ii) *Model Privacy*: the data owners do not learn anything about the FM $\mathcal{M}_\psi$.

**Assumptions**: To simplify the double-blind federated adaptation problem and make it practically feasible, we make the following assumptions: (i) *Auxiliary dataset for preliminary adaptation*: We assume that the LSP has access to an independent auxiliary dataset $\mathcal{D}_{\text{aux}}$, which allows it to perform preliminary adaptation of the given FM into an image classifier. Note that this public dataset may not even correspond to the target image classification task. (ii) *Modularity of FM*: We further assume that the FM has a modular architecture, which can be represented as a sequence of $L$ blocks, i.e., $\mathcal{M}_\psi = \mathcal{B}_{\psi_1} \circ \mathcal{B}_{\psi_2} \cdots \circ \mathcal{B}_{\psi_L}$. Specifically, a transformer architecture is considered in this work. (iii) *Thin client*: The data owners do not have the resources to store the FM (or an encrypted version of it) and perform inference (or encrypted inference) using the FM. However, we assume that the data owners have sufficient computational resources to perform fully homomorphic encryption and decryption operations. (iv) *Powerful server*: The LSP has enough computational resources to perform private inference on encrypted data transmitted by the data owners. Henceforth, we refer to the LSP and data owners as server and clients, respectively. (v) *Semi-honest threat model*: Both the server and the clients are assumed to be semi-honest, i.e., they follow the adaptation protocol honestly, but may attempt to violate the privacy constraints.

**Challenges**: The vanilla federated adaptation (see Appendix A) is not privacy-preserving because it requires computation of $\hat{y}_i^k = \widetilde{\mathcal{M}}_{\widetilde{\psi}}(\mathbf{x}_i^k)$, where the core FM $\mathcal{M}_\psi$ is available only at the server and the local training datasets are available only with the respective clients. Hence, it is essential to design a mechanism for computing $\mathcal{M}_\psi(\mathbf{x}_i^k)$ without violating the data and model privacy constraints. Moreover, the sharing of local updates $\left(\theta_k^{(t)}, \eta_k^{(t)}\right)$ with the server could also potentially leak information about the local datasets (Zhu et al., 2019). Hence, the aggregation step in Eq. 7 must be performed securely without revealing the local updates to the server.

## 3 PROPOSED ADAPTATION FRAMEWORK

The proposed framework for double-blind federated adaptation consists of the following steps: (1) **FHE-friendly Distillation**: Prior to federated adaptation, the server performs an offline distillation of the original FM into a FHE-friendly model using the auxiliary dataset. (2) **Encrypted Inference**: During adaptation, each client encrypts its local data using FHE and sends them to the server. The server and client engage in an interactive protocol to perform inference on the encrypted data. (3) **Local Learning**: Based on the intermediate outputs received from the server, each client locally updates the side network and classification head. (4) **Secure Aggregation**: The server securely aggregates these local updates through a multi-party computation (MPC) protocol with the clients. The overall training workflow of the proposed framework is summarized in Figure 2. At the end of this process, each client receives a copy of the global parameters of the side network and classifier. At the time of inference, the encrypted inference step (Step 2) is repeated. Based on the intermediate outputs received from the server, the client utilizes the global side adapter and classifier to obtain the final class prediction.

### 3.1 FHE-FRIENDLY DISTILLATION

The first step in the proposed framework is to distill the knowledge available in the given FM into an FHE-friendly model. The server performs this step using the auxiliary dataset $\mathcal{D}_{aux}$. Assuming that the given FM follows a modular transformer encoder architecture as shown in Figure 1, we can consider the FM as a sequence of $L$ attention blocks. Let $\mathbf{b}_{\ell-1}$ be the input to the $\ell^{\text{th}}$ attention block $\mathcal{B}_{\psi_\ell}$ and $\mathbf{b}_\ell$ be the corresponding output. The objective of FHE-friendly distillation is to learn an FHE-friendly approximation of $\mathcal{B}_{\psi_\ell}$ denoted as $\widehat{\mathcal{B}}_{\widehat{\psi}_\ell}$ so that when the encrypted input $\mathcal{E}(\mathbf{b}_{\ell-1})$ is provided as input, the server can compute the encrypted output $\mathcal{E}(\mathbf{b}_\ell) = \widehat{\mathcal{B}}_{\widehat{\psi}_\ell}(\mathcal{E}(\mathbf{b}_{\ell-1}))$. Here, $\mathcal{E}$ denotes the encryption operation.

**Approximating Non-linear Functions**: The primary challenge in encrypted inference is the approximation of the non-linear functions because most well-known FHE schemes (e.g., CKKS (Cheon et al., 2017)) allow only polynomial operations. A transformer attention block has three key non-linear operations: Softmax, GELU, and LayerNorm. In this work, Softmax is approximated

using a Taylor series approximation of the exponential function ($e^x$):

$$e^x = \sum_{i=0}^{\infty} \frac{x^i}{i!} \approx \sum_{i=0}^{d} \frac{x^i}{i!}, \tag{1}$$

followed by normalization through division by the sum of the calculated exponential values. The error bound of this approximation is the remainder term, e.g. $\frac{e^\xi}{(d+1)!} x^{d+1}$, for some $\xi$ between 0 and $x$. Furthermore, GELU activation is approximated via a simple quadratic function:

$$\text{GELU}(x) \approx \text{Quad}(x) = 0.125x^2 + 0.25x + 0.5. \tag{2}$$

The LayerNorm function and Softmax require a division step, which is implemented based on the iterative protocol described in (Cheon et al., 2019) following Goldschmidt's algorithm:

$$\frac{1}{x} = \frac{1}{1 - (1 - x)} = \prod_{i=0}^{\infty} \left( 1 + (1 - x)^{2^i} \right) \approx \prod_{i=0}^{d} \left( 1 + (1 - x)^{2^i} \right), \tag{3}$$

where $x \in (0, 2)$.

**Distilling the FM**: After approximating all nonlinearities (Softmax, GELU, and Inverse) in the FM and developing an FHE-friendly FM, we enhance the performance of the FHE-friendly model through knowledge distillation. Initially, we apply layerwise distillation by aligning hidden representations at four positions: (i) the embedding layer, (ii) the attention matrix within each transformer block (considering raw values before normalization), (iii) hidden states after each transformer block, and (iv) the final prediction layer (Jiao et al., 2019; Hinton, 2015; Li et al., 2022). During distillation, we dedicate the first half of the training epochs to distilling the embedding and transformer blocks (both the attention matrix and hidden states) and then proceed to distill the prediction layer, following the approach in (Jiao et al., 2019).

## 3.2 Encrypted Inference

Since FMs often have a very deep neural network architecture, it is challenging to perform encrypted inference over the full FM based on currently available FHE constructs that support only limited multiplicative depth. Attempting to do so will result in large approximation errors, which may be tolerable for simple inference tasks, but can potentially derail the federated learning process. One potential solution is to apply bootstrapping at regular intervals along the computational path. However, bootstrapping is computationally very expensive and makes the framework practically infeasible (especially under the thin client assumption). Hence, we propose performing encrypted inference only over a single transformer attention block at a time. The output of this block is sent back to the client, which decrypts the intermediate result, performs re-encryption of the intermediate result $\mathcal{E}(\mathcal{F}(\mathcal{E}(\mathbf{b}_\ell)))$, and returns it to the server. Here, $\mathcal{F}$ denotes the decryption operation.

The downsides of performing encrypted inference over one attention block at a time are two-fold. Firstly, it increases the communication cost because encrypted intermediate outputs are exchanged between the client and the server after every block in every FL round. Since communication efficiency is not one of our core constraints, we consider the increased communication cost as a limitation, not a red flag. However, since the intermediate representations $\mathbf{b}_\ell$ after every attention block are accessible to the client in plaintext form, a malicious client could use $(\mathbf{b}_{\ell-1}, \mathbf{b}_\ell)$ pairs for multiple training samples to mount a model extraction attack (Liang et al., 2024) and learn the parameters of each transformer block. This clearly violates the model privacy constraint.

To circumvent the above problem and preserve model privacy, we leverage the fact that each communication round in the federated adaptation phase involves a batch of samples. Let $\mathcal{E}(\mathbf{B}_\ell) = [\mathcal{E}(\mathbf{b}_\ell^1), \mathcal{E}(\mathbf{b}_\ell^2), \cdots, \mathcal{E}(\mathbf{b}_\ell^n)]$ be a batch of encrypted intermediate representations corresponding to a client, where $n$ is the batch size. Before sending these representations to the client, the server applies a $n \times n$ permutation matrix $\mathbf{\Pi}_\ell$ and sends only the permuted batch $\mathcal{E}(\mathbf{B}_\ell) \cdot \mathbf{\Pi}_\ell = [\mathcal{E}(\mathbf{b}_\ell^{\pi(1)}), \mathcal{E}(\mathbf{b}_\ell^{\pi(2)}), \cdots, \mathcal{E}(\mathbf{b}_\ell^{\pi(n)})]$ to the client. Here, $[\pi(1), \pi(2), \cdots, \pi(n)]$ represents a random permutation of $[1, 2, \cdots, n]$. This permutation matrix $\mathbf{\Pi}_\ell$ can be randomly selected for each block $\ell$ in each communication round. Thus, the client never sees corresponding pairs $(\mathbf{b}_{\ell-1}^i, \mathbf{b}_\ell^i)$ for any training sample $i$ in the batch, ensuring protection against model extraction attacks.

The overall encrypted inference protocol can be summarized as follows. At the beginning of the collaboration, each client $k$ encrypts (using its public key) each input in its local dataset based on a FHE encryption scheme and sends the encrypted inputs $\{\mathcal{E}(\mathbf{x}_i)\}_{i=1}^{N_k}$ and the corresponding encrypted labels $\{\mathcal{E}(y_i)\}_{i=1}^{N_k}$ to the server. The server applies the embedding function on the encrypted data to obtain the input to the first attention layer $\{\mathcal{E}(\mathbf{b}_0^i)\}_{i=1}^{N_k}$. Subsequently, for each FL round, the server randomly selects a batch of $n$ samples from this set, say $\mathcal{E}(\mathbf{B}_0) = [\mathcal{E}(\mathbf{b}_0^1), \mathcal{E}(\mathbf{b}_0^2), \cdots, \mathcal{E}(\mathbf{b}_0^n)]$, and performs encrypted inference using the FHE-friendly first attention block $\widehat{\mathcal{B}}_{\widehat{\psi}_1}$. The output of this block is permuted using randomly selected $\Pi_1$ before being transmitted to the client. The client decrypts these representations (using its private key), re-encrypts again (using its public key), and

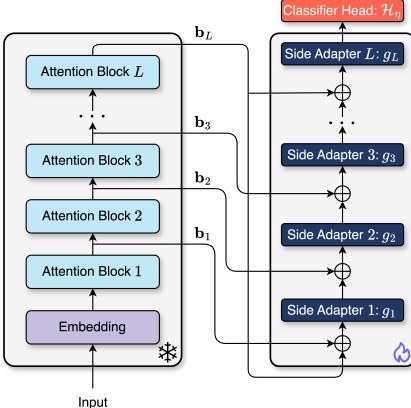

Figure 1: Parallel adapter illustration.

returns it to the server. These re-encrypted values are used as the input to the next attention block and the same encrypted inference process is repeated for all the other blocks $\ell = 2, \cdots, L$.

When the client receives the output of the final transformer attention block $\mathcal{E}(\mathbf{B}_L)$, the decrypted representations are passed through the classification head and the final predictions are again encrypted to get $\mathcal{E}(\hat{\mathbf{Y}}) = [\mathcal{E}(\hat{y}^1), \mathcal{E}(\hat{y}^2), \cdots, \mathcal{E}(\hat{y}^n)]$. These encrypted predictions are sent back to the server for per-sample loss computation in the encrypted domain. The server computes the average (or total) loss over the entire batch and sends this encrypted average loss to the client. The client decrypts this average batch loss and uses it for local learning. During this entire encrypted inference protocol, all the operations on the server are carried out in the encrypted domain. Since the server does not have access to the client's private key, the server learns no information about the client's local data. On the other hand, the client receives a batch of intermediate representations (in plaintext) after each attention block, but samples in each batch are randomly permuted based on $\Pi_\ell$. Since the client cannot access $\Pi_\ell$, it cannot use a model extraction attack to learn any useful information about the FM held by the server, as long as the batch size $n$ is sufficiently large. This ensures that the proposed encrypted inference protocol is double-blind, even though it is interactive in nature.

## 3.3 Local Learning

Though various model adaptation strategies are available, the proposed framework requires an adaptation method that does not require backpropagation of gradients through the FM. This leaves us with only two possible choices: *transfer learning* (where only the classification head is learned) and *parallel adapter* (where both the classification head and a side adapter are learned). In this work, we adopt the low-rank parallel adapter method proposed in (Mercea et al., 2024) (see Figure 1). This method requires access to intermediate representations after every transformer attention block, which is readily available (in plaintext) through the encrypted inference protocol described in Section 3.2.

The output of the low-rank parallel adapter corresponding to the attention block $\ell$ can be expressed as:

$$\mathbf{h}_\ell = g_\ell(\mathbf{b}_\ell + \mathbf{h}_{\ell-1}) + \mathbf{h}_{\ell-1}, \tag{4}$$

where $\mathbf{h}_0 = \mathbf{b}_L$. The adapter function $g_\ell$ is given by:

$$g_\ell(\mathbf{z}) = \alpha \mathbf{W}_\ell^u \mathrm{GELU}(\mathbf{W}_\ell^d \mathbf{z}), \tag{5}$$

where $\mathbf{W}_\ell^d$ and $\mathbf{W}_\ell^u$ are the down and up-projection matrices and $\alpha_\ell$ is the scaling factor at block $\ell$. Since the adapter function in Eq. 5 is applied to each individual sample, the permutation of samples within a batch does not affect this computation. However, the adapter output in Eq. 4 depends on values from two consecutive blocks, which have undergone different permutations. Hence, it is necessary to ensure consistent permutation of the inputs. When operating a batch of samples, Eq. 4 can be reformulated as:

$$\mathbf{H}_\ell = g_\ell(\mathbf{B}_\ell + \mathbf{H}_{\ell-1}) + \mathbf{H}_{\ell-1}, \tag{6}$$

where $\mathbf{H}_0 = \mathbf{B}_L$. Note that the client receives only a permutation of intermediate representations, i.e., $(\mathbf{B}_\ell \cdot \Pi_\ell)$ and not the original $\mathbf{B}_\ell, \forall \ell \in [1, L]$. Hence, to facilitate the computations associated

with the parallel adapter, the server also sends $(\Pi_{\ell-1}^{-1} \cdot \Pi_\ell)$ for all $\ell \in [2, L+1]$ as well as $(\Pi_L^{-1} \cdot \Pi_1)$ to the client. When the client receives $(\mathbf{B}_L \cdot \Pi_L)$, it can compute $\mathbf{H}_0^{'} = (\mathbf{B}_L \cdot \Pi_L) \cdot (\Pi_L^{-1} \cdot \Pi_1) = (\mathbf{B}_L \cdot \Pi_1) = (\mathbf{H}_0 \cdot \Pi_1)$. This can be directly used in Eq. 6 along with $(\mathbf{B}_1 \cdot \Pi_1)$ to compute $\mathbf{H}_1^{'} = (\mathbf{H}_1 \cdot \Pi_2)$. Following the same logic, it is possible to compute $\mathbf{H}_\ell^{'} = (\mathbf{H}_\ell \cdot \Pi_{\ell+1})$, $\forall \ell \in [1, L]$. When the server receives the final encrypted predictions from the client, it can permute the encrypted labels of the batch using $\Pi_{L+1}$ before computing the per-sample losses and aggregating them. It must be emphasized that revealing $(\Pi_{\ell-1}^{-1} \cdot \Pi_\ell)$ for all $\ell \in [2, L+1]$ as well as $(\Pi_L^{-1} \cdot \Pi_1)$ to the client does not leak any information about $\Pi_\ell$ as shown in Proposition 1. Finally, the client locally updates the parallel adapter and classification head in the plaintext domain based on the average loss received from the server employing the same procedure as in (Poirot et al., 2019).

**Proposition 1.** *Let $\mathbf{A}$, $\mathbf{B}$, and $\mathbf{C}$ be $n \times n$ permutation matrices. Given only $\mathbf{A}^{-1}\mathbf{B}$, $\mathbf{B}^{-1}\mathbf{C}$, and $\mathbf{C}^{-1}\mathbf{A}$, it is computationally infeasible to uniquely recover the individual matrices $\mathbf{A}$, $\mathbf{B}$, and $\mathbf{C}$ without additional information.*

*Proof of Proposition 1.* See Appendix C. □

### 3.4 SECURE AGGREGATION

To ensure the secure aggregation of model parameter updates from clients, we leverage secure multi-party computation (MPC) for computing averages in a privacy-preserving manner. This approach, commonly referred to in the literature as Secure Aggregation (Bonawitz et al., 2017), enables the aggregation server to compute the average of client updates without gaining access to the individual updates themselves.

## 4 EXPERIMENTS

Table 1 compares the performance of our method and baselines on the Fed-ISIC2019 dataset with five centers. The auxiliary datasets used at the LSP include (1) Fed-ISIC2019 with only the first center (treated as an in-distribution dataset) and (2) Tiny-ImageNet (treated as an out-of-distribution dataset). The results demonstrate that knowledge transfer from the OOD dataset is effective for all the methods, highlighting that the public dataset can be any available dataset. Notably, our method effectively learns from the OOD FM, with

| Public dataset | Methods | Centralized | Federated |
|---|---|---|---|
| **Fed-ISIC2019 (center=0) (InD)** | Linear probing | 0.6599 | 0.5856 |
| | Full fine-tuning | 0.7811 | 0.6752 |
| | **Ours** | 0.7090 | 0.6679 |
| **Tiny-Imagenet (OOD)** | Linear probing | 0.6372 | 0.5789 |
| | Full fine-tuning | 0.7817 | 0.6985 |
| | **Ours** | 0.7051 | 0.6481 |

Table 1: Performance comparison of our method with baseline approaches on the Fed-ISIC2019 dataset with five clients ($K = 5$), using two auxiliary datasets: Fed-ISIC2019 (center=0) as an in-distribution (InD) dataset and Tiny-ImageNet as an out-of-distribution (OOD) dataset.

a minimal drop compared to the InD FM, highlighting its robustness. We refer the reader to the appendix for dataset details, experimental setup, and additional results and analysis.

## 5 CONCLUSION

This paper has explored the synergistic integration of FL and FM, which offers a promising framework for advancing AI technologies while adhering to data and model privacy and security standards. Our study has particularly concentrated on the efficient fine-tuning of large FMs within a federated framework. We addressed the critical challenges associated with the memory, runtime, and computational demands of deploying large FMs, along with ensuring data privacy, confidentiality, and system security in decentralized environments.

For related works, excluded illustrations, and experiments, refer to Appendix B, D.

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

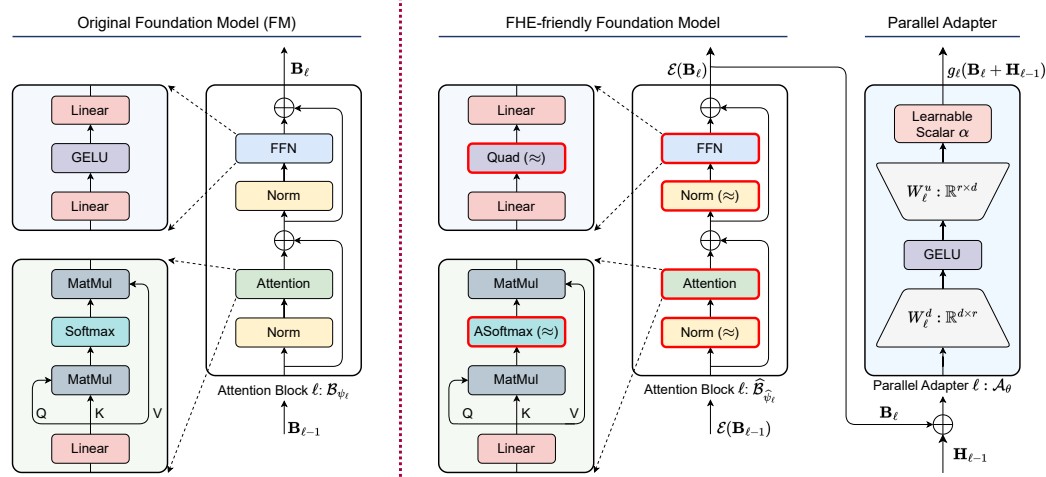

Figure 2: Illustration of the block decomposition and non-linear functions that need to be approximated (highlighted in red).

## A  VANILLA FEDERATED ADAPTATION

Let $\mathcal{L}_k(\theta, \eta) = \frac{1}{N_k} \sum_{i=1}^{N_k} \mathbb{L}(\hat{y}_i^k, y_i^k)$ be the average loss at client $k$, where $\mathbb{L}$ denotes the per-sample loss and $\hat{y}_i^k = \widetilde{\mathcal{M}}_{\widetilde{\psi}}(\mathbf{x}_i^k)$ is the prediction output by the adapted model. Federated adaptation can be posed as a distributed optimization problem (McMahan et al., 2017), where the goal is to learn the global model parameters $(\theta, \eta)$ such that:

$$\min_{(\theta, \eta)} \sum_{k=1}^{K} \alpha_k \mathcal{L}_k(\theta, \eta),$$

where $\alpha_k = \frac{N_k}{\sum_{j=1}^{K} N_j}$. In each round $t$ of FL adaptation, the server broadcasts the previous model parameters $(\theta^{(t-1)}, \eta^{(t-1)})$. Each client computes the local model parameters $(\theta_k^{(t)}, \eta_k^{(t)})$ and these local updates are aggregated by the server to obtain the current global model parameters $(\theta^{(t)}, \eta^{(t)})$. For example, simple FedAvg aggregation function can be represented as follows:

$$\left(\theta^{(t)}, \eta^{(t)}\right) = \sum_{k=1}^{K} \alpha_k \left(\theta_k^{(t)}, \eta_k^{(t)}\right). \tag{7}$$

Note that $t \in [1, T]$, where $T$ is the number of communication rounds and the model parameters $(\theta_k^{(0)}, \eta_k^{(0)})$ for the first round are typically initialized randomly by the server.

## B  RELATED WORK

**Foundation models**: FMs have been highly successful in computer vision (Dosovitskiy et al., 2020; Radford et al., 2021; Liu et al., 2021), natural language processing (Radford et al., 2019; Liu et al., 2019; Yang et al., 2019; He et al., 2020; Clark et al., 2020), and beyond (Sharir et al., 2021; Ranasinghe et al., 2022). In particular, the two-stage training strategy has shown to be effective, where FMs are first pre-trained on a large dataset for general understanding and then fine-tuned on a small downstream dataset to learn task-specific features. However, their vast scale introduces significant challenges, particularly in fine-tuning, that hinder their practical applicability.

**Private inference**: The advent of machine learning as a service (MLaaS) has underscored the critical need for privacy-preserving techniques in ML, particularly in inference tasks. The concept of private inference (PI) has emerged as a pivotal solution to safeguard data and model privacy (Mohassel & Zhang, 2017; Srinivasan et al., 2019; Shen et al., 2022; Tan et al., 2021; Zhang et al.,

2023d; Li et al., 2022; Huang et al., 2022). Vision transformers (ViTs), despite their superior performance in various tasks, pose significant challenges due to their computational complexity, especially when coupled with cryptographic constructs such as FHE (Acar et al., 2018; Gentry, 2009; Cheon et al., 2017) and MPC (Goldreich, 1998; Knott et al., 2021) that are required for PI. The literature surrounding the optimization of PI frameworks primarily focuses on reducing computational and communication overheads while maintaining or improving model accuracy. SAL-ViT (Zhang et al., 2023d) is a framework designed to enhance the efficiency of PI on ViT models. Iron (Hao et al., 2022) introduced secure protocols aimed at optimizing matrix multiplication and several complex non-linear functions integral to transformer models, such as Softmax, GELU activations, and LayerNorm. An alternative approach is to develop PI-friendly transformer architectures and softmax approximations, such as MPCViT (Zeng et al., 2023) and MPCFormer (Li et al., 2022). MPCViT introduces an MPC-friendly adaptation of ViTs that aims to balance the trade-off between accuracy and efficiency during inference in MPC settings. MPCFormer utilizes MPC and knowledge distillation to address latency and reduction in inference quality for transformers.

**Adaptation of foundation models**: The primary issue in adapting FMs is their massive size, making it impractical for individual users or clients with limited computational resources to fine-tune or even store them. Various PEFT techniques such as adapters (Houlsby et al., 2019; Lin et al., 2020), prompt learning (Li & Liang, 2021), low-rank adaptation (LoRA) (Hu et al., 2021), and low-rank side adaptation (Mercea et al., 2024) have been proposed. Numerous variants of LoRA, such as AdaLoRA (Zhang et al., 2023c), Delta-LoRA (Zi et al., 2023), IncreLoRA (Zhang et al., 2023a), and QLoRA (Dettmers et al., 2023) are also available. These methods specifically target the transformer attention blocks that form the core of these models. LoRA (Hu et al., 2021) introduces adjustments to the weight matrices within these blocks, thus enabling efficient fine-tuning with a reduced computational load. However, LoRA necessitates backpropagation through the backbone, thus increasing the total time taken to update the model. Examples of studies exploring PEFT techniques for LLMs with federated learning include (Zhang et al., 2023e) and (Zhang et al., 2023b).

## C  PROOF OF PROPOSITION 1

*Proof of Proposition 1.* Let $\boldsymbol{P} = \boldsymbol{A}^{-1}\boldsymbol{B}$, $\boldsymbol{Q} = \boldsymbol{B}^{-1}\boldsymbol{C}$, and $\boldsymbol{R} = \boldsymbol{C}^{-1}\boldsymbol{A}$, where $\boldsymbol{P}$, $\boldsymbol{Q}$, and $\boldsymbol{R}$ are known products derived from matrices $\boldsymbol{A}$, $\boldsymbol{B}$, and $\boldsymbol{C}$. Each of $\boldsymbol{A}$, $\boldsymbol{B}$, and $\boldsymbol{C}$ is a permutation matrix, defined by:

$$
\begin{aligned}
\boldsymbol{A}, \boldsymbol{B}, \boldsymbol{C} \quad \in \quad & \mathbb{P}_n = \big\{ \boldsymbol{X} \in \{0,1\}^{n \times n} : \\
& \boldsymbol{X}^T \boldsymbol{X} = \boldsymbol{I}, \boldsymbol{X}\boldsymbol{1} = \boldsymbol{1} \big\}
\end{aligned}
\tag{8}
$$

where $\mathbb{P}_n$ denotes the set of $n \times n$ permutation matrices, $\boldsymbol{X}^T$ is the transpose of $\boldsymbol{X}$, $\boldsymbol{1}$ is a vector of ones, and $\boldsymbol{X}^T = \boldsymbol{X}^{-1}$.

**Non-uniqueness of solutions:** To recover $\boldsymbol{A}$, $\boldsymbol{B}$, and $\boldsymbol{C}$ uniquely, we would need the products $\boldsymbol{P}$, $\boldsymbol{Q}$, and $\boldsymbol{R}$ to uniquely determine a single set of matrices $(\boldsymbol{A}, \boldsymbol{B}, \boldsymbol{C})$. However, for any valid solution $(\boldsymbol{A}, \boldsymbol{B}, \boldsymbol{C})$ that satisfies $\boldsymbol{P} = \boldsymbol{A}^{-1}\boldsymbol{B}$, $\boldsymbol{Q} = \boldsymbol{B}^{-1}\boldsymbol{C}$, and $\boldsymbol{R} = \boldsymbol{C}^{-1}\boldsymbol{A}$, we can construct alternative solutions by applying a left-multiplication with any permutation matrix $\boldsymbol{S} \in \mathbb{P}_n$. Define alternative matrices $\boldsymbol{A}' = \boldsymbol{S}\boldsymbol{A}$, $\boldsymbol{B}' = \boldsymbol{S}\boldsymbol{B}$, and $\boldsymbol{C}' = \boldsymbol{S}\boldsymbol{C}$, then:

$$
\begin{aligned}
\boldsymbol{A}'^{-1}\boldsymbol{B}' &= (\boldsymbol{S}\boldsymbol{A})^{-1}(\boldsymbol{S}\boldsymbol{B}) \\
&= \boldsymbol{A}^{-1}\boldsymbol{S}^{-1}\boldsymbol{S}\boldsymbol{B} \\
&= \boldsymbol{A}^{-1}\boldsymbol{B} = \boldsymbol{P},
\end{aligned}
\tag{9}
$$

and similarly, $\boldsymbol{B}'^{-1}\boldsymbol{C}' = \boldsymbol{Q}$ and $\boldsymbol{C}'^{-1}\boldsymbol{A}' = \boldsymbol{R}$. Thus, the products $\boldsymbol{P}$, $\boldsymbol{Q}$, and $\boldsymbol{R}$ are also satisfied by the set $(\boldsymbol{A}', \boldsymbol{B}', \boldsymbol{C}')$. Since there are $n!$ possible choices for $\boldsymbol{S}$, there are $n!$ distinct sets of matrices $(\boldsymbol{A}', \boldsymbol{B}', \boldsymbol{C}')$ that satisfy the products $\boldsymbol{P}$, $\boldsymbol{Q}$, and $\boldsymbol{R}$.

**Brute-force as the optimal attack strategy:** Given the non-uniqueness property above, an attacker aiming to recover $\boldsymbol{A}$, $\boldsymbol{B}$, and $\boldsymbol{C}$ must resort to brute force, testing all possible configurations of $(\boldsymbol{A}, \boldsymbol{B}, \boldsymbol{C})$ that satisfy the products $(\boldsymbol{P}, \boldsymbol{Q}, \boldsymbol{R})$. The total number of combinations of $(\boldsymbol{A}, \boldsymbol{B}, \boldsymbol{C})$ the attacker would need to test is $n!$ and for $n = 16$, we need $16! \approx 2 \cdot 10^{13}$ combinations.

$\square$

# D  EXPERIMENTS AND RESULTS

## D.1  DATASETS

To validate and implement our approach, we will utilize a well-known vision transformer (ViT) pretrained on ImageNet-1K (ViT-Base) as the FM. The public auxiliary datasets for distilling the FHE-friendly FM are:

- **Tiny-Imagenet** (Le & Yang, 2015) is a subset of the larger ImageNet dataset, and it contains 200 different classes, each with 500 images, 100 validation/test images, totaling 120000 images.
- **Fed-ISIC2019** (Terrail et al., 2022) is a multi-class dataset of dermoscopy images containing 23247 images across 8 melanoma classes, with high label imbalance. This dataset provides a valuable opportunity to test our framework within the healthcare domain. For the distillation phase of our experiments, we exclusively use data from client 1.

The private downstream datasets are:

- **CIFAR-10 and CIFAR-100** (Krizhevsky et al., 2009) datasets are standard benchmarks for image classification, containing 60000 color images across 10 and 100 classes, respectively.
- **SVHN** (Netzer et al., 2011) dataset is a benchmark for digit recognition, consisting of over 600000 color images of house numbers. We utilize 73257 images for training and 26032 for testing, disregarding the remainder.
- **Fed-ISIC2019.** The remaining data points for centers 2-6 are used in the fine-tuning experiments, aligning well with the federated setup, as the dataset is tailored for federated learning. For the centralized setup, all data points are pooled into a single client.

## D.2  EXPERIMENTAL SETUP

We employ the Vision Transformer (ViT) (Dosovitskiy et al., 2020), pre-trained on the ImageNet-1k dataset (Russakovsky et al., 2015) (ViT-Base), with a backbone dimension of $384 \times 384$. For the first phase of our framework, obtaining the FHE-friendly FM, we use Adam optimizer with a learning rate of $10^{-4}$ for distilling the transformer blocks for 15 epochs and $10^{-5}$ for distilling the prediction layer for the remaining 15 epochs, totaling 30 epochs. We set the batch size to 16 due to the substantial memory demands. We use MSE loss for the first phase of the distillation and the combination of cross-entropy loss and Kullback-Leibler (KL) divergence losses for the second phase. We set the polynomial order of exponential approximation to 6 and the order of the inverse to 7. For the second phase of our framework, federated adaptation, we use SGD optimizer with a learning rate of 0.001 for Linear probing and our proposed method and $5 \cdot 10^{-5}$ for full fine-tuning

| Datasets | Methods | Centralized | Federated ($K = 5$) | | |
|---|---|---|---|---|---|
| | | Pooled | Dirichlet ($\alpha = 100$) | Dirichlet ($\alpha = 1$) | Dirichlet ($\alpha = 0.01$) |
| **CIFAR-10** | Linear probing | 0.9226 | 0.9203 | 0.9191 | 0.7447 |
| | Full fine-tuning | 0.9635 | 0.9759 | 0.9725 | 0.8857 |
| | **Ours** | 0.9428 | 0.9471 | 0.9413 | 0.8540 |
| **CIFAR-100** | Linear probing | 0.7476 | 0.7486 | 0.7414 | 0.5317 |
| | Full fine-tuning | 0.8361 | 0.8684 | 0.8611 | 0.7882 |
| | **Ours** | 0.7930 | 0.7929 | 0.7808 | 0.6620 |
| **SVHN** | Linear probing | 0.5938 | 0.5879 | 0.5732 | 0.3385 |
| | Full fine-tuning | 0.9680 | 0.9763 | 0.9692 | 0.7601 |
| | **Ours** | 0.9232 | 0.9329 | 0.9249 | 0.7431 |

Table 2: Comparison of the performance of our method against baseline approaches across three datasets (CIFAR-10, CIFAR-100, and SVHN) in both centralized and federated learning settings. The federated experiments involve five clients ($K = 5$) with data partitioned using a Dirichlet distribution at varying levels of heterogeneity ($\alpha = 100, 1, 0.01$).

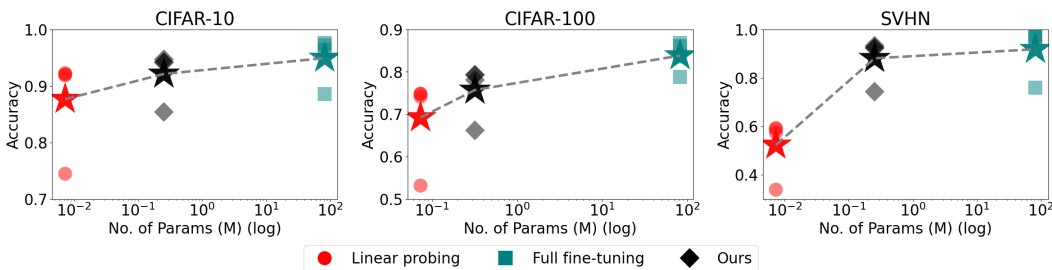

Figure 3: Accuracy vs. no. of trainable parameters trade-off (x-axis in log scale) derived from Table 2, illustrating the performance of the methods across three datasets.

experiments. We set the total number of communication rounds to $T = 50$, and we use a learning rate scheduler with a decay factor of $0.1$ at rounds $[25, 40]$. We set the batch size to $16$ unless otherwise specified. We use cross-entropy loss to evaluate the effectiveness of the global model and report balanced accuracy. We use Dirichlet distribution-based splitting for all our experiments except the Fed-ISIC2019 dataset, which is naturally partitioned. All our experiments are conducted on NVIDIA A100-SXM4-40GB GPUs on an internal cluster server, with each run utilizing a single GPU.

### D.3 RESULTS

**Main results:** In Table 2, the performance of our proposed method and baseline methods across three datasets (CIFAR-10, CIFAR-100, and SVHN) is evaluated under both centralized and federated learning settings. The federated learning experiment uses a Dirichlet partitioning strategy with varying levels of data heterogeneity, controlled by the Dirichlet concentration parameter $\alpha$ (ranging from $100$ to $0.01$). The results demonstrate that full fine-tuning achieves the highest accuracy across all datasets and settings, particularly excelling in more homogeneous federated scenarios, but it is computationally expensive. Linear probing performs well under centralized and homogeneous federated settings. When the task is relatively more straightforward, it fails to work with the SVHN dataset, reaching only $0.5938$ accuracy in the centralized setting, and struggles when there is extreme heterogeneity. Our proposed method delivers robust and competitive results across all settings, showing strong resilience to non-i.i.d data, making it suitable for practical federated learning scenarios (refer to Figure 3).

**Scalability results:** Table 3 illustrates the scalability of our method and the baseline approaches on the CIFAR-10 dataset with increasing number of clients ($N = \{10, 20, 50\}$) using Dirichlet partitioning ($\alpha = 1.0$) and a fixed batch size of 8. Full fine-tuning achieves the highest accuracy for $K = 10$ and $K = 20$ but becomes infeasible for $K = 50$ due to GPU limitations (we use only one GPU for each of our experiments). Linear probing demonstrates stable performance, but our method outperforms linear probing in all the settings, balancing compute efficiency, scalability, and accuracy and demonstrating its suitability for federated setups with a large number of clients.

| Methods | $K = 10$ | $K = 20$ | $K = 50$ |
|---|---|---|---|
| Linear probing | 0.9167 | 0.9142 | 0.9007 |
| Full fine-tuning | 0.9739 | 0.9513 | $N/A$ |
| **Ours** | 0.9446 | 0.9422 | 0.9287 |

Table 3: Scalability analysis of the proposed method to baseline approaches on the CIFAR-10, with varying number of clients $K \in \{10, 20, 50\}$ under a Dirichlet concentration parameter of $1.0$ for data partitioning. $N/A -$ one GPU is insufficient to run the experiment.

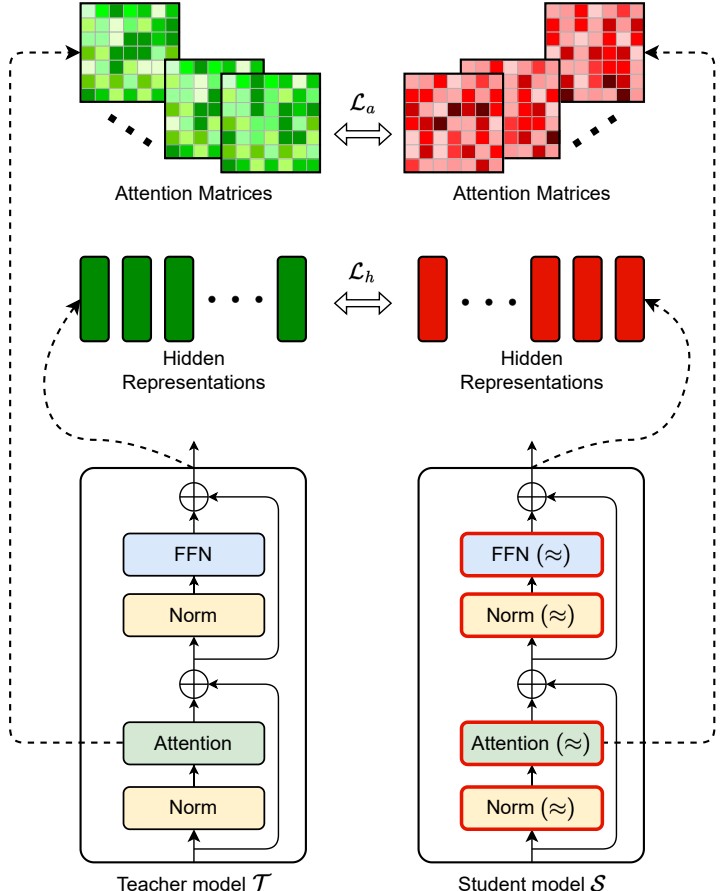

Figure 4: Schematic illustration of the layer-wise knowledge distillation (Equations 12 and 13).

# E  HOW DOES THE DISTILLATION WORK?

We adopt the transformer distillation approach (Jiao et al., 2019), that consists of two key phases:
(i) transformer-layer distillation and (ii) prediction-layer distillation. For generality, let the original
transformer model be referred to as the teacher model $(\mathcal{T})$ and the approximation-enabled trans-
former model as the student model $(\mathcal{S})$. We assume that both models share the same architecture
but differ only in their non-linear components (Softmax, GELU, LayerNorm). Further, we outline
the primary sub-layers of the transformer blocks:

- **Attention.** The attention function is formulated as follows:

$$\mathbf{A} = \frac{\mathbf{Q}\mathbf{K}^{T}}{\sqrt{d_k}}, \tag{10}$$

  followed by a softmax operation. We are specifically interested in an un-normalized atten-
  tion matrix $\mathbf{A}$.

- **Feed-forward network** is formulated as follows:

$$\mathbf{H}(x) = \text{GELU}(x\boldsymbol{W}_1 + b_1)\boldsymbol{W}_2 + b_2. \tag{11}$$

In the first stage of distillation, we perform an attention-based distillation (Eq. 10) and hidden
representations based distillation (Eq. 11). More precisely,

$$\mathcal{L}_a = \frac{1}{h}\sum_{i=1}^{h}\left\|\mathbf{A}_i^{\mathcal{S}} - \mathbf{A}_i^{\mathcal{T}}\right\|^2, \tag{12}$$

where $h$ is the number of attention heads. And, the distillation of the hidden representation is formulated as follows:

$$\mathcal{L}_h = \|\mathbf{H}^{\mathcal{S}} - \mathbf{H}^{\mathcal{T}}\|^2, \tag{13}$$

where the matrices $\mathbf{H}^{\mathcal{S}}$ and $\mathbf{H}^{\mathcal{T}}$ are hidden representations of the respective models. For a detailed look, please refer to Figure 4, which illustrates how layer-wise distillation works. Then, the total loss of the first stage is simply defined as:

$$\mathcal{L} = \mathcal{L}_a + \mathcal{L}_h. \tag{14}$$

Further, we perform a prediction-layer distillation following the knowledge distillation approach of (Hinton, 2015) by matching logits and using the following objective function:

$$\mathcal{L}_p = \mathcal{L}_{CE}(\boldsymbol{z}^{\mathcal{S}}/\tau, \boldsymbol{z}^{\mathcal{T}}/\tau), \tag{15}$$

where $\boldsymbol{z}^{\mathcal{S}}$ and $\boldsymbol{z}^{\mathcal{T}}$ are logit vectors produced by student and teacher models, respectively. $\mathcal{L}_{CE}$ is a cross-entropy loss and $\tau$ is a temperature parameter to produce softer probability distributions over classes. We set $\tau$ to 5 in all our experiments. Finally, the final loss objective is defined as:

$$\mathcal{L} = \begin{cases} \mathcal{L}_a + \mathcal{L}_h & \triangleright \text{ Stage I,} \\ \mathcal{L}_p & \triangleright \text{ Stage II.} \end{cases} \tag{16}$$

As outlined in the main paper, the total number of epochs is set to 30, with 15 epochs allocated to each stage.

## F    EXPERIMENTAL RESULTS

### F.1    MAIN RESULTS

Table 4 provides additional details on the number of trainable parameters, the latency, and the GPU memory required to execute the respective experiments across the given datasets and methods. These results complement Tables 2 and 1. As shown in the table, our proposed method closely matches Linear Probing in terms of the number of parameters and the latency required to process a single sample, while full fine-tuning exhibits a significant disparity in effectiveness. Additionally, the GPU memory required for our method is comparable to that of Linear Probing, whereas full fine-tuning demands nearly twice as much memory.

### F.2    DISTILLATION RESULTS

Following the findings discussed in Section E, we present performance plots for two key processes: (i) fine-tuning the teacher model $\mathcal{T}$ on a public auxiliary dataset $\mathcal{D}_{aux}$, and (ii) distilling the student model $\mathcal{S}$ on the same dataset $\mathcal{D}_{aux}$ using the trained teacher model $\mathcal{T}$. Figure 5 illustrates

| Datasets | Methods | No. of params (M) | | Latency (ms) | | Memory (GB) |
|---|---|---|---|---|---|---|
| **CIFAR-10 / SVHN** | Linear probing | 0.0073 | $(1.00 \times)$ | 15.1 | $(1.00 \times)$ | 9.39 |
| | Full fine-tuning | 82.1096 | $(11247.89 \times)$ | 47.4 | $(3.14 \times)$ | 18.13 |
| | **Ours** | 0.2536 | $(34.74 \times)$ | 15.7 | $(1.04 \times)$ | 9.08 |
| **CIFAR-100** | Linear probing | 0.0733 | $(1.00 \times)$ | 15.7 | $(1.00 \times)$ | 9.40 |
| | Full fine-tuning | 82.1756 | $(1121.09 \times)$ | 47.4 | $(3.03 \times)$ | 18.13 |
| | **Ours** | 0.3196 | $(4.36 \times)$ | 16.3 | $(1.04 \times)$ | 9.08 |
| **Fed-ISIC2019** | Linear probing | 0.0059 | $(1.00 \times)$ | 20.2 | $(1.00 \times)$ | 8.71 |
| | Full fine-tuning | 82.1082 | $(13916.64 \times)$ | 51.2 | $(2.54 \times)$ | 17.32 |
| | **Ours** | 0.2522 | $(42.75 \times)$ | 21.1 | $(1.05 \times)$ | 8.39 |

Table 4: Comparison of the efficiency of our method with baseline approaches in terms of the number of parameters, latency and the memory requirement across four datasets (CIFAR-10/SVHN, CIFAR-100, and Fed-ISIC2019). Latency is measured per data point and includes both forward and backward passes. The last column (Memory (GB)) indicates the GPU memory required to conduct the experiment.

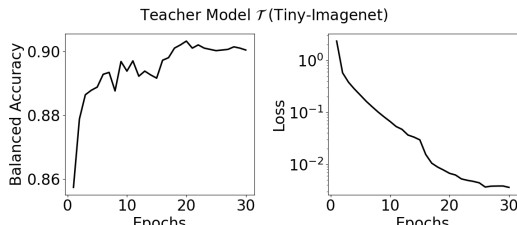

Figure 5: Performance plot of fine-tuning the teacher model on the public auxiliary dataset $\mathcal{D}_{aux}$ (Tiny-Imagenet). The plot on the left shows the balanced accuracy, while the plot on the right presents the loss performance (in a logarithmic scale).

Figure 6: Performance plot of the distillation process on the Tiny-Imagenet dataset. The plot on the left depicts the balanced accuracy across both stages, while the right plot shows the loss performance, with the dashed line indicating the beginning of Stage 2 distillation (Eq. 16).

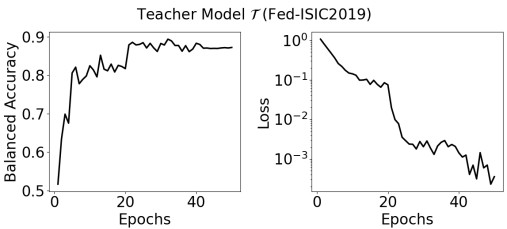

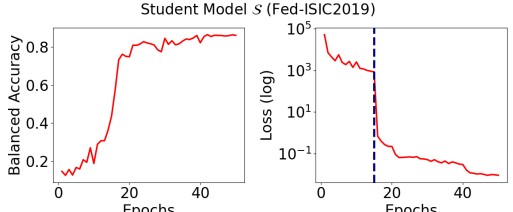

Figure 7: Performance plot of fine-tuning the teacher model on the Fed-ISIC2019 dataset with center=0. The plot on the left shows the balanced accuracy, while the plot on the right presents the loss performance (in a logarithmic scale).

Figure 8: Performance plot of the distillation process on the Fed-ISIC2019 dataset with center=0. The plot on the left depicts the balanced accuracy across both stages, while the right plot shows the loss performance, with the dashed line indicating the beginning of Stage 2 distillation (Eq. 16).

the performance metrics − balanced accuracy and loss − of the fine-tuning process on the Tiny-ImageNet dataset. After completing this training, we proceed with distillation for the approximation-enabled student model. Figure 6 shows the performance plot for the distillation process on the Tiny-ImageNet dataset. In the loss behavior plot (on the right), a dashed line marks the start of the second stage. During the first stage, as described in Eq. 16, transformer-layer distillation is performed (showing higher values due to the high-dimensional attention and hidden representation matrices). Then, the second-stage distillation follows, which is a prediction-layer distillation (Hinton, 2015). The learning rate scheduler is employed at epochs 15 and 25 with a decay factor of 0.1. Figures 7 and 8 present analogous performance plots for the Fed-ISIC2019 dataset, using center=0 as the public auxiliary dataset $\mathcal{D}_{aux}$. The learning rate scheduler is employed at epochs 20 and 40 with a decay factor of 0.1.

### F.3 APPROXIMATION RESULTS

In this section, we revisit and elaborate on the approximations introduced in the main paper.

**Softmax.** Given a vector $\boldsymbol{z} \in \mathbb{R}^n$ with components $z_i$, the softmax function is defined as:

$$\text{softmax}(z_i) = \frac{e^{z_i}}{\sum_{j=1}^{n} e^{z_j}}$$

where $e^{z_i}$ denotes the exponential function applied to the component $z_i$. The goal is to approximate the softmax function using a polynomial approximation of the exponential function, $P_n(\boldsymbol{z}_i)$, and to estimate the maximum deviation in softmax values due to this approximation.

| Degree $d$ | Memory (GB) | Latency (s) | Attention loss $(\mathcal{L}_a)$ | Representation loss $(\mathcal{L}_h)$ | One-epoch accuracy |
|---|---|---|---|---|---|
| 2 | 16.82 | 0.41 | 44.37 | 915.46 | 75.17 |
| 4 | 19.68 | 0.48 | 38.31 | 748.62 | 83.85 |
| 6 | 22.54 | 0.52 | 33.92 | 655.26 | 84.36 |
| 8 | 25.39 | 0.58 | 30.91 | 603.53 | 84.47 |
| 10 | 28.99 | 0.62 | 28.17 | 573.54 | 84.16 |
| 16 | 36.83 | 0.81 | 20.31 | 527.95 | 85.08 |
| $\infty$ (True) | 15.36 | 0.35 | 13.59 | 508.64 | 86.91 |

Table 5: **Softmax approximation results.** Latency is computed per a batch of samples, when batch size is set to 8. One-epoch accuracy refers to the accuracy we obtain when performing one full pass over the data for only one epoch. This experiment is carried on the Fed-ISIC2019(center=0) dataset.

The Taylor series provides a polynomial approximation of $e^x$ around $x = 0$ up to $d$ terms:

$$e^x \approx P_d(x) = \sum_{k=0}^{d} \frac{x^k}{k!}$$

The error bound of this approximation is given by the remainder term $R_n(x)$, expressed as:

$$R_d(x) = \frac{e^\xi}{(d+1)!} x^{d+1}$$

for some $\xi$ between 0 and $x$, indicating the error in approximating $e^x$ with $P_d(x)$.

**Softmax approximation results.** Table 5 presents softmax approximations while keeping all other non-linear components fixed. Using Eq. 1, we vary the polynomial degree $d$ for the approximation during a single epoch of knowledge distillation (Section E, Stage I) and compare the results to the true softmax. The table reports key performance metrics, including attention loss $\mathcal{L}_a$ (Eq. 12), representation loss $\mathcal{L}_h$ (Eq. 13), and accuracy. Additionally, it provides efficiency metrics such as the GPU memory usage and latency for processing a batch of samples. Balancing the trade-offs between time, memory requirements, and performance drop, we chose $d = 6$ for all experiments.

| Degree $d$ | Memory (GB) | Latency (ms) | Attention loss $(\mathcal{L}_a)$ | Representation loss $(\mathcal{L}_h)$ | One-epoch accuracy |
|---|---|---|---|---|---|
| 2 | 22.42 | 118.70 | 61.50 | 1698.19 | 57.87 |
| 4 | 22.44 | 119.64 | 57.42 | 1534.97 | 76.64 |
| 6 | 22.45 | 120.27 | 45.67 | 1352.91 | 81.77 |
| 7 | 22.46 | 120.43 | 44.11 | 1334.32 | 82.96 |
| 8 | 22.47 | 120.97 | 45.10 | 1352.55 | 83.23 |
| 10 | 22.48 | 121.70 | 43.23 | 1332.62 | 80.61 |
| 12 | 22.50 | 122.98 | 40.40 | 1302.41 | 82.55 |
| 16 | 22.53 | 124.76 | 38.86 | 1292.36 | 83.04 |
| $\infty$ (True) | 22.46 | 10.57 | 38.77 | 1289.71 | 83.15 |

Table 6: **Approximation of the reciprocal (inverse function).** Latency is computed per a batch of samples (aggregate over $L$ transformer blocks), when batch size is set to 8. One-epoch accuracy refers to the accuracy we obtain when performing one full pass over the data for only one epoch. This experiment is carried on the Fed-ISIC2019 (center=0) dataset.

**Inverse.** Since homomorphic encryption supports only addition and multiplication operations, it cannot perform division directly. To make it work in the encryption domain, we approximate the inverse function. The details of this approximation are presented in Eq. 3, and the corresponding algorithm is outlined in Algorithm 1. To ensure that the value of $x$ falls within the range $(0, 2)$, we determine the maximum possible value of $x$ in the plaintext domain and scale it accordingly in the encrypted domain using this maximum value.

---

**Algorithm 1** Inverse algorithm

---

**Input:** $0 < x < 2$, degree $d \in \mathbb{N}$
**Output:** an approximation of $1/x$ (Eq. 3)

1: $a_0 \leftarrow 2 - x$
2: $b_0 \leftarrow 1 - x$
3: **for** $n \leftarrow 0$ to $d - 1$ **do**
4:     $b_{n+1} \leftarrow b_n^2$
5:     $a_{n+1} \leftarrow a_n \cdot (1 + b_{n+1})$
6: **end for**
7: **return** $a_d$

---

**Inverse approximation results.** Table 6 summarizes the results of inverse function approximations, keeping all other non-linear components fixed and utilizing the softmax approximation with a 6th-degree polynomial. Following Algorithm 1 and Eq. 3, we varied the polynomial degree $d$ under the same conditions described above. While the memory and time requirements remain nearly identical across different values of $d$, the focus is on performance metrics. Based on these considerations, we selected $d = 7$, as it closely approximates the true inverse in terms of losses and accuracy.

