# OpenReview forum: "A Framework for Double-Blind Federated Adaptation of Foundation Models"
_ICLR.cc/2025/Workshop/MCDC — MCDC @ ICLR 2025_

### Official Review · Reviewer_7JNg · 2025-02-25

**Rating:** 6
**Confidence:** 3
**Fit:** 5

**Summary:**

This paper introduces a double-blind federated adaptation framework for foundation models (FMs), ensuring that data privacy and model privacy are preserved simultaneously. The primary challenge addressed is adapting large FMs for downstream tasks when data is distributed across multiple entities and cannot be centralized due to privacy regulations, while the FM owner (Learning Service Provider - LSP) cannot share the model due to proprietary concerns.

The proposed framework leverages Fully Homomorphic Encryption (FHE) and Secure Multi-Party Computation (MPC) to enable encrypted inference and privacy-preserving training. The FM is first distilled into a sequence of FHE-friendly blocks, enabling inference without exposing model parameters. Adaptation is achieved using low-rank parallel adapters that do not require backpropagation through the FM, reducing computational costs. A privacy-preserving permutation scheme prevents clients from extracting model information, and secure aggregation ensures the confidentiality of model updates.

Empirical results on four datasets (CIFAR-10, CIFAR-100, SVHN, and Fed-ISIC2019) demonstrate that the proposed approach achieves competitive accuracy while ensuring privacy. Compared to full fine-tuning and linear probing, the method offers a balance between efficiency, scalability, and privacy, making it suitable for real-world federated learning applications. The paper provides a theoretically sound and practically feasible solution for privacy-preserving FM adaptation in decentralized settings.

**Reason For Giving A Higher Score:**

The paper presents a strong and novel contribution to the field of federated learning by addressing the challenge of adapting foundation models (FMs) in a double-blind setting, ensuring both model and data privacy. The proposed approach leverages Fully Homomorphic Encryption (FHE) and Secure Multi-Party Computation (MPC) to enable privacy-preserving inference and adaptation without exposing sensitive data or model parameters. The methodology is well-founded, introducing low-rank parallel adapters to avoid the computational overhead of full fine-tuning while maintaining strong adaptation performance. The empirical validation across multiple datasets (CIFAR-10, CIFAR-100, SVHN, and Fed-ISIC2019) demonstrates the method’s effectiveness, showing that it outperforms linear probing while being more efficient than full fine-tuning. Additionally, the framework remains scalable under varying levels of data heterogeneity, making it suitable for real-world federated learning applications. The introduction of a privacy-preserving permutation scheme further strengthens security by mitigating model extraction attacks. While the computational overhead of FHE and the lack of evaluation on larger-scale foundation models present areas for improvement, the balance between privacy, efficiency, and accuracy justifies a high score.

**Reason For Giving A Lower Score:**

One major concern is the computational overhead of Fully Homomorphic Encryption (FHE), which, while ensuring strong privacy guarantees, is known to be computationally expensive. The paper does not provide a detailed analysis of the practical latency and efficiency trade-offs, which raises concerns about its feasibility for large-scale deployment. Additionally, the communication overhead caused by frequent encrypted exchanges between the server and clients is not thoroughly analyzed, potentially making the approach impractical for real-world federated learning settings with limited bandwidth. Another limitation is the lack of evaluation on large-scale foundation models—the experiments focus on relatively small datasets (CIFAR-10, CIFAR-100, SVHN, and Fed-ISIC2019), but do not assess scalability to more complex models. Furthermore, the paper does not include a robustness analysis against adversarial settings, such as clients sending incorrect updates or attempting to infer model parameters through side-channel attacks. Without such evaluations, the security claims of the framework remain incomplete. Lastly, while the proposed privacy-preserving permutation scheme helps defend against model extraction attacks, its effectiveness is not empirically tested against known adversarial techniques. Given these shortcomings, particularly regarding scalability, computational cost, and robustness analysis, the paper’s contributions, while valuable, may require further refinement before being fully applicable to large-scale federated learning scenarios.

**Strengths And Weaknesses:**

Strength:

The framework avoids full fine-tuning, which is computationally expensive, by using low-rank parallel adapters that do not require backpropagation through the FM.
The FHE-friendly model decomposition enables efficient inference while maintaining privacy.

Empirical results on CIFAR-10, CIFAR-100, SVHN, and Fed-ISIC2019 demonstrate that the method scales well across different datasets.
The approach is tested under varying degrees of data heterogeneity (Dirichlet partitioning).

The method outperforms linear probing in most settings while being more efficient than full fine-tuning.
Shows resilience to non-i.i.d. data, making it suitable for real-world federated learning (FL) applications.

The privacy-preserving permutation scheme prevents clients from reconstructing FM parameters, mitigating potential model extraction attacks.

Weaknesses:

While FHE ensures strong privacy, it is computationally expensive. The paper does not discuss practical latency implications in detail, which could be a concern for large-scale deployments.
Bootstrapping operations in FHE are computationally demanding, potentially limiting real-time applications.

The framework is evaluated on relatively small datasets (CIFAR-10, CIFAR-100, SVHN, and Fed-ISIC2019).
No evaluation on larger FMs that are typically used in foundation model applications.

The framework requires frequent encrypted intermediate data exchanges between the server and clients, which may result in high communication costs in real-world federated settings.
While the authors acknowledge this limitation, further analysis is needed to quantify the impact.

The paper does not evaluate how the framework performs under adversarial attacks, such as malicious clients sending incorrect updates or inference-time perturbations.
Additional robustness experiments could strengthen the paper’s claims on secure adaptation.

**Suggestions:**

Optimize FHE Computation: Explore techniques such as quantized FHE, hybrid encryption schemes, or alternative secure computation methods to reduce computational cost.

Evaluate on Larger Models: Test the approach on larger-scale FMs to demonstrate scalability.

Quantify Communication Overhead: Include a detailed communication cost analysis to assess the feasibility of deploying the framework in real-world federated settings.

Security Testing Against Adversaries: Conduct robustness experiments to evaluate resistance to model inversion attacks, gradient manipulation attacks, or adversarial perturbations.

Overall, the paper presents a strong and novel approach for privacy-preserving FM adaptation but could be further improved with additional large-scale evaluations and security analyses.

---

### Official Review · Reviewer_bxzT · 2025-03-03

**Rating:** 6
**Confidence:** 2
**Fit:** 4

**Summary:**

The authors propose a method for adapting foundation models to downstream tasks in a federated learning framework that ensures "double-blind" privacy, meaning the model owner never reveals the model parameters, and the data owners do not expose their raw data. Their approach first distills the foundation model into a version that is compatible with fully homomorphic encryption, enabling secure encrypted computations. Then, they train low-rank parallel adapters that efficiently fine-tune the distilled model for specific tasks without compromising privacy.

**Reason For Giving A Higher Score:**

I am not deeply familiar with the related work in this area, so it is hard to judge how novel it is to integrate techniques like distillation and fully homomorphic encryption into a single framework.

**Reason For Giving A Lower Score:**

Similarly, since I am not an expert in each of the frameworks used (distillation, FHE, ..) I couldn't fully assess the soundness of how the authors combined these techniques.

**Strengths And Weaknesses:**

Strengths:

- The paper tackles an important problem: adapting the increasing number of foundation models to local settings using FL while ensuring privacy for both model and data.
- The paper is very comprehensive, detailing each component of the approach.
- The work combines very modern techniques (distillation, FHE, MPC, ) and discusses practical issues (e.g. mitigating malicious clients) along with potential solutions.

Weaknesses:

- The structure of the paper is strange; it seems to have been shortened from a longer version without fully ensuring a coherent result. For example, no experiments/results are presented in the main text, even though they are listed as main contributions.
- Similarly, for a non-domain expert it is not clear which methods are new and which are simply applied. For instance, half a page in the main text explains the vanilla FL setting and FedAVG. This could be moved to the appendix to allow space to highlight novel frameworks and results.
- The writing is dense, with very long sentences and paragraphs that make it hard to identify the key concepts (like the "encrypted inference" and "local learning" sections).

**Suggestions:**

To address the weaknesses, I would suggest restructuring the paper so that the key ideas and results fit within six pages. Move detailed descriptions of established methods to the appendix, allowing the main text to focus on the novel contributions. The description of the new approach should be more concise and clearly highlight the critical elements. Additionally, it is essential to include at least some experimental results in the main text so readers can properly evaluate the effectiveness of the approach

---

### Decision · Program_Chairs · 2025-03-06

**Decision:**

Accept

**Comment:**

This paper introduces a method for double-blind FL to fine-tune  foundation models without sharing model parameters or data.  The paper is relevant to the topic of decentralized training. Overall, we recommend acceptance and suggest the authors to take reviewers' comments in consideration.